# Effect of Chicken Egg White-Derived Peptide and Hydrolysates on Abnormal Skin Pigmentation during Wound Recovery

**DOI:** 10.3390/molecules28010092

**Published:** 2022-12-22

**Authors:** Pei-Gee Yap, Chee-Yuen Gan, Idanawati Naharudin, Tin-Wui Wong

**Affiliations:** 1Analytical Biochemistry Research Centre (ABrC), Universiti Sains Malaysia, University Innovation Incubator Building, SAINS@USM Campus, Lebuh Bukit Jambul, Bayan Lepas 11900, Penang, Malaysia; 2Non-Destructive Biomedical and Pharmaceutical Research Centre, Smart Manufacturing Research Institute, Universiti Teknologi MARA Selangor, Puncak Alam 42300, Selangor, Malaysia; 3Particle Design Research Group, Faculty of Pharmacy, Universiti Teknologi MARA Selangor, Puncak Alam 42300, Selangor, Malaysia

**Keywords:** bioactive peptide, egg white protein, melanogenesis, skin lightening agent, skin hyperpigmentation

## Abstract

Abnormal skin pigmentation commonly occurs during the wound healing process due to the overproduction of melanin. Chicken egg white (CEW) has long been used to improve skin health. Previous published works had found CEW proteins house bioactive peptides that inhibit tyrosinase, the key enzyme of melanogenesis. The current study aimed to evaluate the anti-pigmentation potential and mechanism of the CEW-derived peptide (GYSLGNWVCAAK) and hydrolysates (CEWH_mono_ and CEWH_di_), using a cell-based model. All of these peptide and hydrolysates inhibited intracellular tyrosinase activity and melanin level up to 45.39 ± 1.31 and 70.01 ± 1.00%, respectively. GYSLGNWVCAAK and CEWH_di_ reduced intracellular cAMP levels by 13.38 ± 3.65 and 14.55 ± 2.82%, respectively; however, CEWH_mono_ did not affect cAMP level. Moreover, the hydrolysates downregulated the mRNA expression of melanogenesis-related genes, such as *Mitf*, *Tyr*, *Trp-1* and *Trp-2*, but GYSLGNWVCAAK only suppressed *Tyr* gene expression. Downregulation of the genes may lower the catalytic activities and/or affect the structural stability of TYR, TRP-1 and TRP-2; thus, impeding melanogenesis to cause an anti-pigmentation effect in the cell. Outcomes from the current study could serve as the starting point to understand the underlying complex, multifaceted melanogenesis regulatory mechanism at the cellular level.

## 1. Introduction

Post-inflammatory hyperpigmentation (PIH) is a common acquired hyperpigmentation disorder during the wound healing process as a sequela of cutaneous injury, infection or inflammatory reactions. The hallmark of PIH includes the appearance of flat, dark brown and sometimes blue-grey patches or spots on the wound after its recovery. It can happen to all skin types but tends to affect dark skinned individuals (Fitzpatrick type IV, V and VI, where the skin colour corresponds to olive, brown and black/dark brown, respectively) with greater frequency and severity [1]. Although the exact aetiology of PIH is unclear, hyperpigmentation is associated with the overproduction or deposition of melanin [2,3]. Tyrosinase is the major enzyme in melanogenesis that catalyses the hydroxylation of tyrosine and the oxidation of ʟ-DOPA to *o*-dopaquinone [4]. The *o*-dopaquinone undergoes a series of redox reactions to form dopachrome, where it is then decarboxylated to 5,6-dihydroxyindole (DHI) or enzymatically rearranged into 5,6-dihydroxyindole-2-carboxylic acid (DHICA) by tyrosinase-related protein 2 (TRP-2). DHI and DHICA are polymerised into eumelanin in the presence of tyrosinase-related protein 1 (TRP-1). Both TRP-1 and TRP-2 also function as structural proteins to stabilize tyrosinase and melanosome structures during melanogenesis [5].

Since the overactivity of tyrosinase can induce the overproduction of melanin, inhibition of this enzyme has become the major target to treat or prevent abnormal skin hyperpigmentation [6]. In fact, the global skin lightening product market is expected to reach approximately USD 16 billion by 2030 with a consumer preference shift towards natural, organic-based products [7]. This is because long-term usage of chemical-based products may pose health and safety concerns to consumers, such as cytotoxicity, erythema or dermatitis [8,9], especially when the regulation of skincare products is not as strict compared to prescribed drugs. Therefore, a compound capable of inhibiting tyrosinase activity sourced from natural, organic materials with minimum side effects may favour consumer demand and could be explored for its potential in skin lightening formulations.

Chicken egg white (CEW), apart from being a food source, is a popular ingredient in home skincare remedies. Egg white proteins are rich in bioactive components known as bioactive peptides. Peptides are small, inactive protein fragments encrypted within a parent protein molecule [10]. Peptide-based tyrosinase inhibitors have been reported from various food sources [11,12]. According to the previous studies by our research group, CEW hydrolysates, i.e., 10 mg/mL of CEWH_mono_ and CEWH_di_ inhibited 45.3 ± 0.66 monophenolase and 48.1 ± 0.23% diphenolase activities of mushroom tyrosinase, respectively [13]. To compare with other reports, the collagen hydrolysate showed 15.44–30.20% tyrosinase inhibition at 5 mg/mL [14]. The jellyfish hydrolysates showed 50% inhibition of tyrosinase activity at 14.1–24.5 mg/mL [15]. The CEWH_di_-derived peptide, GYSLGNWVCAAK, also gave an IC_50_ value of 3.04 ± 0.39 mM for tyrosinase diphenolase activity [16], which is more potent than that of FPY [17] and IQSPHFF [18], with IC_50_ values of 3.22 ± 0.9 and 4.0 mM, respectively. It is of particular interest to further investigate the intracellular anti-pigmentation effect and mechanism of the CEW-derived hydrolysates and peptide in order to evaluate their potential as a skin lightening agent against PIH. Therefore, this work is a follow-up study aimed at determining the effects of GYSLGNWVCAAK, CEWH_mono_ and CEWH_di_ on intracellular tyrosinase activity, melanin level, and cAMP level, and their regulatory role in the melanogenesis signaling pathway using the B16F10 melanoma cell line.

## 2. Results and Discussion

### 2.1. Culturing of B16F10 Melanoma Cells

The B16F10 melanoma cell line is commonly used for pigmentation-related studies. This is because human and mouse tyrosinases are highly homologous with 86 and 84% identity at the protein and cDNA levels, respectively [19]. The cell line also shares a similar regulatory mechanism of melanin synthesis with human melanocytes. In addition, the cells can firmly attach to the culture flask and establish a homogenous cell population; thus, allowing microscopic view of cellular differentiation in terms of morphology and melanogenesis [20,21]. The growth morphology of B16F10 melanoma cells is shown in Figure 1a. Upon thawing (Day 0), viable cells with intact cell membranes appeared round and shiny with a hollow ring under a microscope. The cells attached to the culture flask and gradually established their characteristic epithelial-like and spindle shape (Day 1–5). The cells were sub-cultured on Day 6 as the cells had reached >90% confluency. Intracellular melanin production was observed after the third sub-culture (Figure 1b).

### 2.2. Cell Cytotoxicity Assay

To investigate the cytotoxicity effect of peptide and CEW hydrolysates, B16F10 melanoma cells were administered with various concentrations of GYSLGNWVCAAK, CEWH_mono_ and CEWH_di_ and then subjected to MTT colorimetric assay. According to the ISO 10993-5:2009 guideline, a cellular treatment was considered toxic should it induce a >30% reduction of cell viability [22]. Based on Figure 2, the viability of the cells treated with GYSLGNWVCAAK, CEWH_mono_ and CEWH_di_ were maintained above 75% at 5–100 µg/mL, suggesting an unlikely cytotoxic effect of the samples. In other reports, the milk and rice bran protein-derived peptides even promoted the growth of B16F10 cells at 100–500 µM [23,24]. On the other hand, cell viability decreased significantly (*p* < 0.05) for cells treated with 1000 µg/mL CEWH_mono_ (20%) and CEWH_di_ (30%) compared to the untreated control. For treatment using the positive control, cell viability dropped beyond 75% at concentrations >10 µg/mL. This cytotoxicity effect could be due to the auto-oxidation of EGCG into hydrogen peroxide and superoxide anion in culture media, which induces oxidative stress to the cells [25]. In short, 100 µg/mL was used as the maximum treatment concentration for GYSLGNWVCAAK, CEWH_mono_ and CEWH_di_ and the results were compared with EGCG at lower levels of 5 and 10 µg/mL in the subsequent analyses. 

### 2.3. Intracellular Tyrosinase Activity

The tyrosinase activity in B16F10 melanoma cells treated by GYSLGNWVCAAK and CEWH_di_ is shown in Figure 3. To the best of our knowledge, there were no reports on the determination of monophenolase activity using tyrosinase extracted from cell models. This is because it is necessary to first accumulate ʟ-DOPA through diphenolase activity to allow tyrosinase to reach a steady state for full activation of its catalytic activity. Since tyrosinase exists in three oxidation forms, each showing a different ability to catalyze phenol or catechol substrates [26], the kinetics of the monophenolase activity is complicated and difficult to monitor [27]. Therefore, the monophenolase reaction was not determined in the current study. 

GYSLGNWVCAAK and CEWH_di_ significantly (*p* < 0.05) suppressed the diphenolase activity of B16F10 cells compared to the untreated control in a dose-independent manner. At 10 µg/mL, GYSLGNWVCAAK and CEWH_di_ gave the highest activity suppression (45.39 ± 1.31 and 50.89 ± 0.36%, respectively), which were higher than the positive control EGCG (39.65 ± 1.16%) at an equivalent concentration. For 10 µg/mL of CEWH_di_ treatment, the effect was comparable to 100 µg/mL of cod skin hydrolysate and 752.4 µg/mL of milkfish scale hydrolysate, which inhibited 48.93 and 50% of intracellular tyrosinase activities, respectively [28,29]. The effect of 10 µg/mL (7.88 µM) GYSLGNWVCAAK treatment was higher than that of 350 µM crocodile blood-derived peptides [30]. The inhibitory effect of GYSLGNWVCAAK was ascribed to its direct interactions with tyrosinase and copper-chelating ability, as previously discussed [16]. However, a significant lower inhibition of tyrosinase activity was observed at treatment concentrations >10 µg/mL. The dose-independent inhibition of tyrosinase activity could be due to a different progression between tyrosinase activity and melanin polymerization [31]. To elaborate, the optimal pH of tyrosinase was pH 6.5–7.0 [32], whereas melanin polymerization could be accelerated at alkaline conditions [31]. Since GYSLGNWVCAAK is a basic peptide (isoelectric point, pI = pH 8.77), increased concentration of the peptide could raise melanosomal microenvironment pH to reduce tyrosinase activity, but at the same time, promote melanin polymerization. The basic side chain of lysine in GYSLGNWVCAAK could act as proton acceptor and centre of clustering for the negatively charged melanin as the polymerization of melanin monomers involves deprotonation [31]. Previous works have shown that basic proteins or peptides can enhance melanin production in vitro [31,33]. Therefore, melanin polymerization may be a more prominent step progressing at a higher rate than tyrosinase catalysis at a higher peptide concentration, triggering higher tyrosinase activity at >10 µg/mL of peptide. 

### 2.4. Intracellular Melanin Level

The intracellular melanin level of B16F10 melanoma cells treated by various concentrations of GYSLGNWVCAAK, CEWH_mono_, CEWH_di_ and EGCG is shown in Figure 4. All treatments significantly (*p* < 0.05) lowered the intracellular melanin level compared to the untreated control and the trend of inhibition corresponded to that of intracellular tyrosinase activity (Section 2.3). The intracellular melanin level of B16F10 cells treated with GYSLGNWVCAAK showed a positive correlation, to some extent (*p* < 0.10), with the intracellular tyrosinase activity (r = 0.845). The strongest inhibition of intracellular melanin level (70.01 ± 1.00%) was recorded in B16F10 cells treated with 10 µg/mL of GYSLGNWVCAAK (Figure 4a). At the same treatment concentration, the inhibition of peptide was more profound than that of EGCG (40.8 ± 2.32%; Figure 4d). Moreover, this inhibitory effect was >2-fold stronger than that of crocodile blood-derived peptides [30]. However, at higher treatment concentrations (50 and 100 µg/mL), GYSLGNWVCAAK induced weaker inhibition on melanin synthesis (34.01 ± 1.41 and 23.55 ± 1.51%, respectively). Melanin polymerization may be enhanced by the basic side chain of peptide, as suggested previously (Section 2.3). 

For CEWH_mono_ (Figure 4b) and CEWH_di_ (Figure 4c), the intracellular melanin level was the lowest (69.99 ± 1.31 and 47.55 ± 0.24%, respectively) at a treatment concentration of 10 µg/mL. The anti-pigmentation effects of CEWH_mono_ and CEWH_di_ were relatively higher than other reported hydrolysates from cod skin and sea cucumber gelatine [28,34]. A significant (*p* < 0.05) positive correlation was found between the intracellular tyrosinase activity and melanin level of B16F10 cells treated with CEWH_di_ (r = 0.951). Since the pH of CEWH_mono_ and CEWH_di_ were 7.95 ± 0.1 and 8.02 ± 0.01, respectively, the basic property of hydrolysates may favour melanin polymerization, as previously discussed (Section 2.3). In another study, B16F10 cells treated with 0.050, 0.100, 0.210 and 0.638 µg/mL chicken feather meal hydrolysate resulted in approximately 0, 21, 14 and 5% inhibition of melanin synthesis, respectively [35]. Higher suppression of melanin production (i.e., lower intracellular melanin level) at low treatment concentration, which is similar to the current result, was observed. However, the authors neither determined the pH of hydrolysate nor accounted the effect of pH on intracellular melanin synthesis. Instead, the authors found no correlation between tyrosinase activity and melanin content as the hydrolysate inhibited the former in a dose-dependent manner but not in the latter. Although this scenario was not observed in the current study, it was noteworthy that melanin inhibition may not always be a direct consequence of tyrosinase inhibition since melanogenesis is a multi-step reaction involving various regulatory players. Based on the results from Section 2.2, Section 2.3 and Section 2.4, cellular treatments with 10 µg/mL of GYSLGNWVCAAK, CEWH_mono_ and CEWH_di_ produced the strongest inhibition of intracellular tyrosinase activity and melanin level without significant cell cytotoxicity. Therefore, 10 µg/mL of cellular treatment was used in the following analyses.

### 2.5. Intracellular cAMP Level

The intracellular cAMP level of B16F10 melanoma cells treated with 10 µg/mL of GYSLGNWVCAAK, CEWH_mono_, CEWH_di_ and EGCG is shown in Figure 5. Compared to the untreated control group, GYSLGNWVCAAK, CEWH_di_ and EGCG significantly reduced the cAMP level by 13.38 ± 3.65 (*p* < 0.05), 14.55 ± 2.82 (*p* < 0.05) and 16.04 ± 5.46% (*p* < 0.01), respectively. The inhibition caused by GYSLGNWVCAAK was slightly lower than that of zebrafish phosvitin-derived peptide Pt5, which reduced 19.0% of cAMP level at 10 µg/mL [36]. In another report, a 48.77% decrement of cAMP level was recorded in B16F10 cells treated with 1.6 mg/mL (2.8 mM) of grass carp scale collagen peptide FTGML [37]. Oyster hydrolysate was also found to reduce cAMP level by approximately 30% at a treatment concentration of 100 µg/mL [38]. An elevated level of intracellular cAMP was associated with upregulation of melanogenesis through the cAMP and/or phosphatidylinositol 3-kinase (PI3K) pathways. Therefore, the current results suggest that GYSLGNWVCAAK, CEWH_di_ and EGCG treatments may impair melanogenesis by lowering the intracellular cAMP level. The cAMP level of B16F10 cells treated with 10 µg/mL of CEWH_mono_, on the contrary, did not show significant difference (*p* > 0.05) with that of the control group. This indicated that the inhibitory effect of CEWH_mono_ on the intracellular melanin level (Section 2.4) was independent of cAMP. 

### 2.6. RT-qPCR Analysis

The effect of cellular treatment with 10 µg/mL of GYSLGNWVCAAK, CEWH_mono_, CEWH_di_ and EGCG on the mRNA expression level of the *Mitf*, *Tyr*, *Trp-1* and *Trp-2* genes normalized to the untreated control is shown in Table 1. All sample treatments decreased the mRNA expression level of the genes except GYSLGNWVCAAK. The positive control, EGCG, also downregulated all genes similar to sesamol and pyruvic acid [39,40]. Interestingly, treatment of cells with 10 µg/mL of GYSLGNWVCAAK did not alter the mRNA expression of *Mitf* but downregulated the *Tyr* gene with 0.76-fold and upregulated *Trp-1* and *Trp-2* gene with 1.03 and 1.52-fold, respectively. The inconsistency of the results suggests that GYSLGNWVCAAK regulated the tyrosinase family genes independent of each other as well as *Mitf*, as opposed to most reported anti-melanogenesis peptides. Considering that GYSLGNWVCAAK significantly (*p* < 0.05) reduced intracellular tyrosinase activity and the melanin level (Section 2.3 and Section 2.4), the peptide was postulated to suppress melanogenesis at the post-translational level instead of at the transcriptional level. For instance, pleiotrophin-treated melanocytes showed reduced protein expression of MITF and tyrosinase but did not affect the mRNA level of the *Mitf* gene. Pleiotrophin was later found to activate the mitogen-activated protein kinase (MAPK) signalling pathway to induce MITF protein degradation through ubiquitination [41]. For *Trp-1* and *Trp-2* genes, they could be regulated irrespective of *Mitf* and *Tyr* genes [42,43]. Although it is unclear why the genes were regulated independently, the fact that GYSLGNWVCAAK reduced the intracellular cAMP level of B16F10 cells (Section 2.5) could trigger several antagonistic molecular events, i.e., reduced transcriptional activity due to suppression of MITF through the PKA or PI3K signalling pathways, and increased MITF degradation through the MAPK signalling pathway. In short, further investigation is necessary to figure out the anti-melanogenesis mode of action of GYSLGNWVCAAK. 

For the hydrolysates, CEWH_mono_ and CEWH_di_ suppressed *Mitf*, *Tyr*, *Trp-1* and *Trp-2* genes. Downregulation of the genes by CEWH_di_ could be cAMP-dependent but not for CEWH_mono_ because the hydrolysate did not significantly (*p* > 0.05) reduce the intracellular cAMP level (Section 2.5). CEWH_mono_ was postulated to act independently of cAMP by directly interacting with the members of the PKA, PI3K and/or MAPK signalling pathways. For instance, the fermented microalgae-derived peptide MGRY promoted the activation of ERK in the MAPK signalling pathway, which induced MITF ubiquitination and degradation [44], whereas sesamol advocated the phosphorylation of Akt and GSK3β in the PI3K/Akt/GSK3β pathway to suppress MITF activity [39]. Besides, CEWH_mono_ could act through the Wingless-related integration site (Wnt)/β-catenin signalling pathway since *Mitf* is a downstream target gene of Wnt [45]. Secreted frizzled-related protein 5 (SFRP5) and cardamonin were among the reported anti-pigmentation agents that downregulated melanogenesis-related genes via this pathway without involving cAMP [46]. Inhibition of *Mitf* suppressed the expression of melanogenesis-related genes and, hence, hindered the production of melanin. 

## 3. Materials and Methods

### 3.1. Materials

Peptide GYSLGNWVCAAK (95.0% purity) was synthesized and purchased from Genscript Biotech Corp., Singapore, whereas the CEW-derived hydrolysates, CEWH_mono_ and CEWH_di_ were prepared according to Yap and Gan [13]. The B16F10 melanoma cell line (ATC.CRL-6475) was purchased from the American Type Culture Collection (ATCC), United States. All chemicals and reagents used in this study were of analytical grade.

### 3.2. Cell Culture and Maintenance of Cell Line

To revive the frozen cell line, the cells were quickly thawed in a 37 °C water bath and seeded in a T25 cell culture flask containing 8 mL of prewarmed complete growth medium (CGM). CGM contained Dulbecco’s Modified Eagle Medium (DMEM) fortified with 10% foetal bovine serum (FBS) and 1% Penicillin-Streptomycin (PenStrep). The cells were maintained in a 5% CO_2_ incubator (MMM Medcenter Einrichtungen GmbH, München, Germany) at 37 °C. The cells were observed under a microscope to check for successful cell attachment and morphological changes. Sub-culturing or passaging of cells was performed once they were grown to approximately 90% confluency and the sub-cultivation ratio used was 1:9. Briefly, the medium was discarded, and the cells were rinsed using 2 mL phosphate-buffered saline (PBS) to remove traces of serum containing protease inhibitors. Then, 1 mL of trypsin–EDTA solution was loaded into the flask and left undisturbed for a few minutes. Cell detachment was checked using a microscope. In total, 2 mL of CGM was added to inhibit the activity of the trypsin–EDTA. The cell suspension was centrifuged at 1200 rpm for 5 min and the supernatant was discarded. For the sub-culture, a pellet of cells was resuspended with an adequate amount of CGM. In total, 1 mL of cells was seeded into each T75 culture flask containing 15 mL prewarmed CGM and then maintained in a 5% CO_2_ incubator. To ensure sufficient cell stock for the subsequent analyses, cells from early passages were cryopreserved and 40 cryovials were kept. For cryopreservation, the harvested cell pellet was resuspended in cold, freshly prepared CGM containing 5% DMSO and loaded into a cryovial. The cryovial was kept at −4 °C for 1 h and then at −80 °C for 24 h after which it was kept in a liquid nitrogen tank for long-term storage.

### 3.3. Cytotoxicity Assay

The cytotoxicity effect of the samples on B16F10 melanoma cells was evaluated through the MTT assay [47]. Briefly, 10^5^ cells per well were seeded in a 96-well plate and incubated for 24 h for cell attachment. The culture media were removed and the cells were rinsed with 200 µL PBS. Next, 100 µL of GYSLGNWVCAAK, CEWH_mono_ and CEWH_di_ (5, 10, 50, 100, 500 and 1000 µg/mL) dissolved in culture media were fed to the cells and then further maintained for 48 h. The treatment was discarded and the cells were rinsed with 200 µL PBS. Then, 20 µL MTT solution (5 mg/mL) was added followed by incubation for 4 h at 37 °C. After incubation, the MTT solution was gently withdrawn without disturbing the MTT formazan precipitate. To dissolve the precipitate, 100 µL of DMSO was loaded with gentle shaking at 100 rpm for 15 min. The absorbance was measured at 570 nm using an ELISA reader (Infinite M200, Tecan, Männedorf, Switzerland). EGCG was used as the positive control. The cell viability was calculated using the equation:(1)$$Cell\;viability\;\left(\%\right)=\frac{A_{sample}}{A_{control}}\times 100$$
where *A_control_* was the absorbance of MTT formazan formed without sample treatment, whereas *A_sample_
*was the absorbance of MTT formazan formed after sample treatment.

### 3.4. Determination of Intracellular Tyrosinase Activity

The effect of GYSLGNWVCAAK, CEWH_mono_ and CEWH_di_ on the intracellular tyrosinase activity of B16F10 melanoma cells was assessed using the protocol of Kim et al. [48] with slight modifications. Briefly, 10^5^ cells per well were seeded in a 6-well plate and incubated for 24 h to allow cell attachment. The culture media were discarded, then the cells were rinsed with 200 µL PBS. After that, 2 mL of sample (5, 10, 50 and 100 µg/mL) dissolved in culture media was administered and the cells were further incubated for 48 h. The treatment was removed and the cells were rinsed with 200 µL PBS. The cells were collected by trypsinization and the pellet was rinsed again with PBS. To determine the intracellular tyrosinase activity, the pellet was disrupted using 300 µL 20 mM potassium phosphate buffer (pH 6.8) containing 1% Triton X-100 to release tyrosinase from the melanosomes. The cell lysate was kept at −80 °C for 1 h and thawed at room temperature. Then, the lysate was centrifuged at 12,000 rpm for 30 min at 4 °C to collect the supernatant containing the cellular extracts. In a 96-well plate, 20 µL of ʟ-DOPA (2 mg/mL) was added to 80 µL of supernatant and incubated for 1 h at 37 °C. The absorbance was monitored at 475 nm. EGCG was used as the positive control. The intracellular tyrosinase activity was calculated using the following equation:(2)$$Intracellular\;tyrosinase\;activity\;\left(\%\right)=\frac{A_{sample}}{A_{control}}\times 100$$where *A_control_* was the absorbance of dopachrome formed without sample treatment, whereas *A_sample_
*was the absorbance of dopachrome formed after sample treatment. 

### 3.5. Determination of Intracellular Melanin Level

The cells were treated as previously mentioned (Section 2.4). To determine the intracellular melanin level, the pellet was dissolved in 200 µL of 1 M NaOH containing 10% DMSO and heated at 80 °C for 1 h, as previously described by Zhang et al. [49] The absorbance was then monitored at 405 nm using an ELISA reader. EGCG was used as the positive control. The intracellular melanin level was calculated using the following equation:(3)$$Intracellular\;melanin\;level\;\left(\%\right)=\frac{A_{sample}}{A_{control}}\times 100$$
where *A_control_* was the absorbance of melanin without sample treatment whereas *A_sample_
*was the absorbance of melanin after sample treatment. 

### 3.6. Determination of Intracellular Cyclic Adenosine 3′,5′-Monophosphate (cAMP) Level

The cells were treated as above (Section 2.4) and the pellet was subjected to a freeze-thaw cycle for cell lysis. The cells were then centrifuged at 2000 rpm and 4 °C for 20 min. Supernatant was collected for the detection of the intracellular cAMP level using a cAMP ELISA kit (Bioassay Technology Laboratory, Yangpu, Shanghai, China) as per the guidelines provided by the manufacturer. EGCG was used as the positive control.

### 3.7. Quantitative Reverse Transcription Polymerase Chain Reaction (RT-qPCR)

In a T25 culture flask, 5 × 10^5^ cells were seeded and treated as described above (Section 2.4). The collected cell pellet was immediately stored in −80 °C. Total RNA was extracted using a NucleoSpin^®^ RNA Kit (Macherey-Nagel, Düren, Germany) according to the manufacturer’s instructions. The quality and quantity of the extracted RNA were checked using a Nanodrop 2000c spectrophotometer (Thermo Scientific, Waltham, MA, USA). Reverse transcription of RNA into cDNA was performed using a OneScript^®^ Hot cDNA Synthesis Kit (abm Inc., Richmond, BC, Canada), whereas the RT-qPCR was conducted using BlasTaq™ 2X qPCR MasterMix (abm Inc.). The primer sequences, forward (F) and reverse (R), for *Mitf*, *Tyr*, *Trp-1* and *Trp-2* genes are listed in Table 2.

The genes of interest were amplified according to the following thermocycling conditions: one cycle of 95 °C for 3 min (DNA polymerase activation) and 40 cycles each at 95 °C for 15 s (cDNA denaturation) and 60 °C for 1 min (annealing/extension). The relative fold change in gene expression was calculated according to the Livak method of 2^−(ΔΔ*Ct*)^ in which a fold change of =, < and >1.0 indicates equal, downregulation and upregulation of gene expression compared to the untreated control group, respectively [50]. The cycle threshold (C*_t_*) value was first normalized to that of the reference gene, *Gapdh* to obtain ΔC*_t_* and then normalized to the C*_t_* of the untreated control to obtain ΔΔC*_t_* using the following equations: (4)$$\Delta C_{t\left(sample\right)}=C_{t\left(sample\right)}-C_{t\left(Gapdh\right)}$$
(5)$$\Delta C_{t\left(control\right)}=C_{t\left(control\right)}-C_{t\left(Gapdh\right)}$$
(6)$$\Delta \Delta C_{t}=\Delta C_{t\left(sample\right)}-\Delta C_{t\left(control\right)}$$
(7)$$Fold\;change=2^{-\left(\Delta \Delta C_{t}\right)}$$
where $C_{t\left(sample\right)}$ represents the C*_t_* value after sample treatment; $C_{t\left(control\right)}$ represents the C*_t_* value without sample treatment; and $C_{t\left(Gapdh\right)}$ represents the C*_t_* value of the reference gene. The result was expressed as a range, which incorporated the standard deviation (sd) of ΔΔC*_t_* into the fold change calculation as follows:(8)$$Range\;of\;fold\;change=2^{-\left(\Delta \Delta C_{t}\pm sd\right)}$$

### 3.8. Statistical Analysis

Statistical analyses were conducted using IBM SPSS Statistics version 20.0 (SPSS Institute, Chicago, IL, USA). The results were analysed using one-way ANOVA followed by Duncan’s post hoc test (Section 2.3, Section 2.4 and Section 2.5). Pearson’s correlation test was also performed to evaluate the strength of the correlation between intracellular tyrosinase activity and melanin level (Section 2.4 and Section 2.5). Dunnett’s post hoc test was conducted to compare the means from the experimental groups against the untreated control group (Section 2.6). A *p*-value of <0.05 implied a significant difference between the sample’s means. Each experimental run was performed in triplicate and the result was reported as the mean ± standard deviation.

## 4. Conclusions

To summarize the results, GYSLGNWVCAAK, CEWH_mono_ and CEWH_di_ could reduce the intracellular tyrosinase activity and melanin level through the regulation of cAMP level or via the expression of melanogenesis-related genes. Outcomes from the current study could serve as the starting point to understand the underlying complex, multifaceted melanogenesis regulatory mechanism at the cellular level. A comparison of the results using a normal human skin melanocyte model is necessary to justify the anti-pigmentation effect considering that the samples may exhibit opposing effects on human melanocytes [51]. Nonetheless, the cell permeability test and melanosome uptake assay should be conducted to verify the permeability of peptides and hydrolysates. Western blotting analysis could be conducted to demonstrate the translation of melanogenesis regulatory proteins. The regulatory effects of the samples on protein translation, protein degradation, impairment of melanosome maturation or even cell autophagy [52,53,54] should also be investigated before moving onto animal model or clinical studies. 

## Figures and Tables

**Figure 1 molecules-28-00092-f001:**
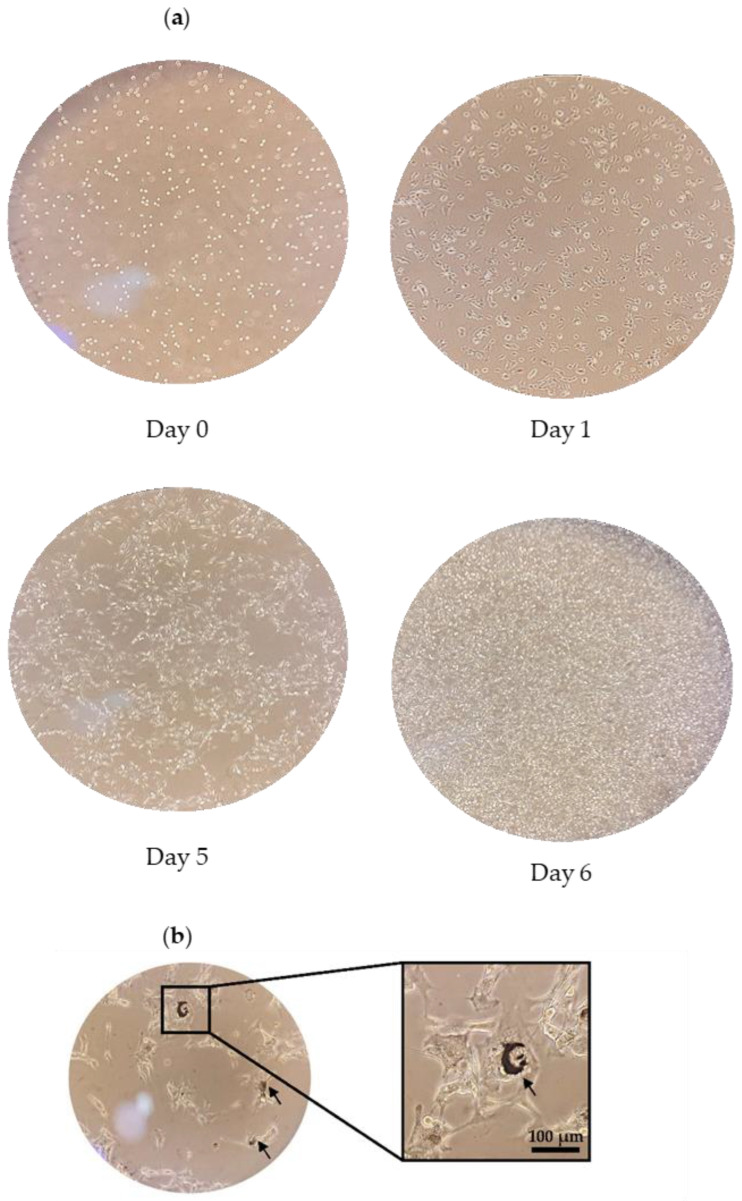
The (**a**) growth morphology of B16F10 melanoma cells viewed under 100× magnification and (**b**) melanin deposition (black arrows) viewed under 200× magnification.

**Figure 2 molecules-28-00092-f002:**
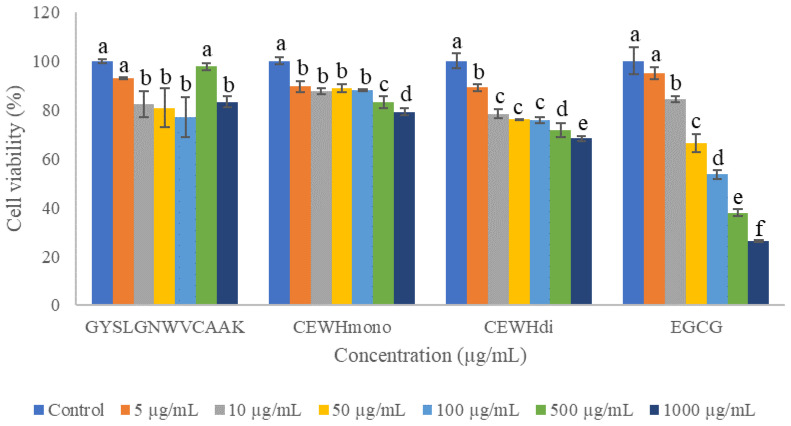
Effects of various concentrations of GYSLGNWVCAAK, CEWH_mono_, CEWH_di_ and EGCG on the cell viability of B16F10 melanoma cells. Note: Results were reported as the mean ± standard deviation (*n* = 3); different lowercase letters above the bars indicate significant difference at *p* < 0.05 according to Duncan’s test.

**Figure 3 molecules-28-00092-f003:**
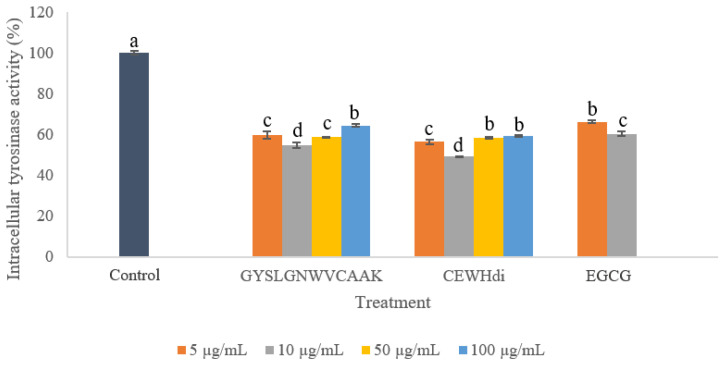
Effects of various concentrations of GYSLGNWVCAAK, CEWH_di_ and EGCG on the intracellular tyrosinase activity of B16F10 melanoma cells. Note: Results were reported as the mean ± standard deviation (*n* = 3); different lowercase letters above the bars indicate significant difference at *p* < 0.05 according to Duncan’s test.

**Figure 4 molecules-28-00092-f004:**
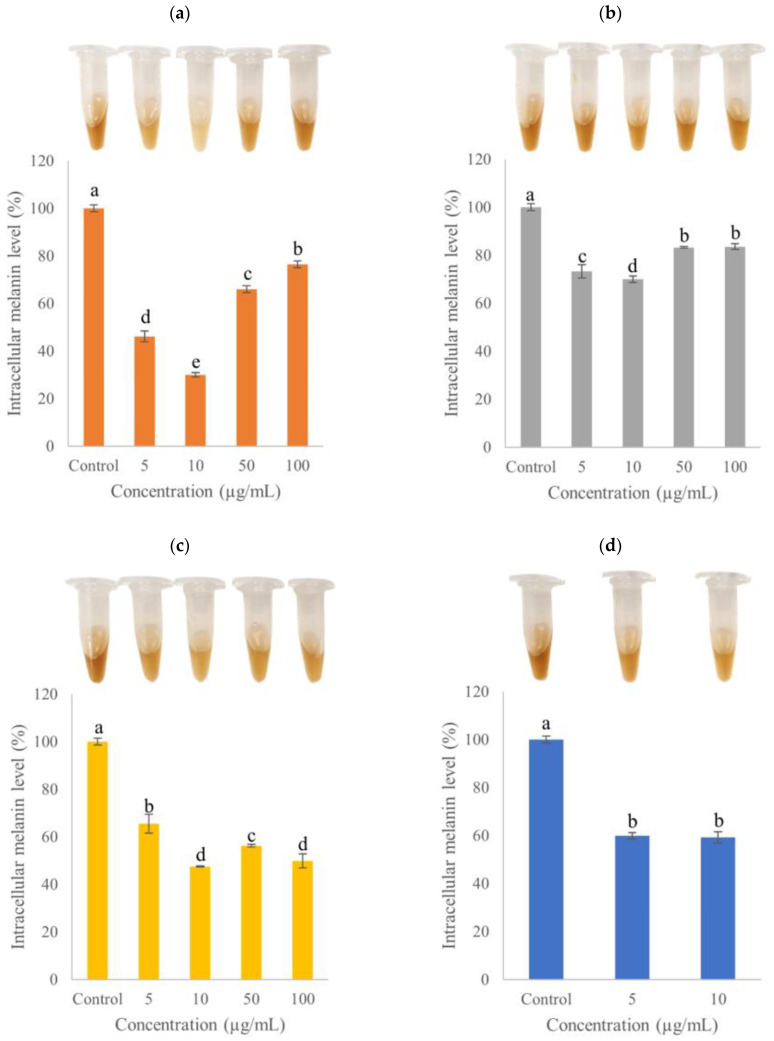
Effects of various concentrations of (**a**) GYSLGNWVCAAK, (**b**) CEWH_mono_, (**c**) CEWH_di_ and (**d**) EGCG on the intracellular melanin level of B16F10 melanoma cells. Note: Results were reported as the mean ± standard deviation (*n* = 3); different lowercase letters above the bars indicate significant difference at *p* < 0.05 according to Duncan’s test.

**Figure 5 molecules-28-00092-f005:**
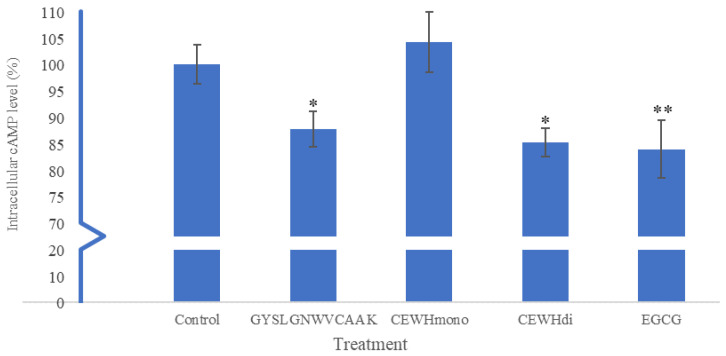
Effects of 10 µg/mL of GYSLGNWVCAAK, CEWH_mono_, CEWH_di_ and EGCG on the intracellular cAMP level of B16F10 melanoma cells. Note: Results were reported as the mean ± standard deviation (*n* = 3); * indicates *p* < 0.05 and ** indicates *p* < 0.01, which were significantly different to the untreated control group according to Dunnett’s test.

**Table 1 molecules-28-00092-t001:** Fold change expression of *Mitf*, *Tyr*, *Trp-1* and *Trp-2* genes after treatment with 10 µg/mL of GYSLGNWVCAAK, CEWH_mono_, CEWH_di_ and EGCG in B16F10 melanoma cells.

Sample	Fold Changes			
	*Mitf*	*Tyr*	*Trp-1*	*Trp-2*
Control	1.00 (0.88–1.14)	1.00 (0.88–1.13)	1.00 (0.78–1.27)	1.00 (0.88–1.14)
GYSLGNWVCAAK	1.00 (0.97–1.03)	0.76 (0.73–0.80)	1.03 (0.97–1.11)	1.52 (1.48–1.57)
CEWH_mono_	0.63 (0.59–0.68)	0.60 (0.59–0.61)	0.42 (0.39–0.46)	0.71 (0.68–0.75)
CEWH_di_	0.88 (0.82–0.95)	0.93 (0.88–0.99)	0.89 (0.88–0.90)	0.71 (0.65–0.80)
EGCG	0.67 (0.60–0.74)	0.58 (0.48–0.71)	0.70 (0.61–0.82)	0.74 (0.59–0.94)

Note: Results were reported as the mean (*n* = 3). The range of fold change is given in the brackets.

**Table 2 molecules-28-00092-t002:** Primer sequences of genes used in the RT-qPCR.

Gene	Primer Sequence	Accession Number
*Mitf*	F: 5′-TACAGAAAGTAGAGGGAGGAGGACTAAG-3′	NM_008601.3
	R: 5′-CACAGTTGGAGTTAAGAGTGAGCATAGCC-3′	
*Tyr*	F: 5′-TTGCCACTTCATGTCATCATAGAATATT-3′	NM_011661.5
	R: 5′-TTTATCAAAGGTGTGACTGCTATACAAAT-3′	
*Trp-1*	F: 5′-GCTGCAGGAGCCTTCTTTCTC-3′	NM_031202.3
	R: 5′-AAGACGCTGCACTGCTGGTCT-3′	
*Trp-2*	F: 5′-GGATGACCGTGAGCAATGGCC-3′	NM_010024.3
	R: 5′-CGGTTGTGACCAATGGGTGCC-3′	
*Gapdh*	F: 5′-GCATCTCCCTCACAATTTCCA-3′	NM_008084.3
	R: 5′-GTGCAGCGAACTTTATTGATGG-3′	

## Data Availability

The data related to this research is included in the results section.

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
