# Peer review of "Effect of Chicken Egg White-Derived Peptide and Hydrolysates on Abnormal Skin Pigmentation during Wound Recovery"

_molecules, 2022, doi:10.3390/molecules28010092_

Round 1
Reviewer 1 Report
This is a manuscript about chicken egg white (CEW) proteins and how they could inhibit tyrosinase, the key enzyme of melanogenesis during wound healing. Specifically, this study evaluates the anti-pigmentation potential and mechanism of the CEW-derived peptide (GYSLGNWVCAAK) and hydrolysates (CEWHmono
and CEWHdi), using a cell-based model. The authors show that these peptides and hydrolysates inhibit intracellular tyrosinase activity and melanin. They also showed that GYSLGNWVCAAK and CEWHdi reduce intracellular cAMP level, while CEWHmono did not . Moreover, the hydrolysates down-regulated mRNA expression of melanogenesis-related genes, such as Mitf, Tyr, Trp-1 and Trp-2.
GYSLGNWVCAAK only suppressed Tyr gene expression. These results suggest that these peptides have a role in the anti-pigmentation effect in the cell.
Overall this is a well presented manuscript.
Author Response
This is a manuscript about chicken egg white (CEW) proteins and how they could inhibit tyrosinase, the key enzyme of melanogenesis during wound healing. Specifically, this study evaluates the anti-pigmentation potential and mechanism of the CEW-derived peptide (GYSLGNWVCAAK) and hydrolysates (CEWHmono and CEWHdi), using a cell-based model. The authors show that these peptides and hydrolysates inhibit intracellular tyrosinase activity and melanin. They also showed that GYSLGNWVCAAK and CEWHdi reduce intracellular cAMP level, while CEWHmono did not . Moreover, the hydrolysates down-regulated mRNA expression of melanogenesis-related genes, such as Mitf, Tyr, Trp-1 and Trp-2.
GYSLGNWVCAAK only suppressed Tyr gene expression. These results suggest that these peptides have a role in the anti-pigmentation effect in the cell.
Overall this is a well presented manuscript.
Response: Thank you for accepting the manuscript for publication.
Reviewer 2 Report
Line 116-120: the authors should correct that adding L-dopa to measure monophenolase activity is not an artefact, since for the enzyme to reach steady state in its action it must accumulate L-dopa in the medium. On the other hand, it is not necessary to measure monophenolase activity.
To study the potency of these peptides, the IC50 value could be determined for example with the mushroom enzyme.
Author Response
Line 116-120: the authors should correct that adding L-dopa to measure monophenolase activity is not an artefact, since for the enzyme to reach steady state in its action it must accumulate L-dopa in the medium. On the other hand, it is not necessary to measure monophenolase activity.
Response: Thank you for pointing out the misleading statement. The statement has been revised to: “This is because it is necessary to first accumulate ÊŸ-DOPA through the diphenolase activity to allow tyrosinase to reach steady state for full activation of its catalytic activity. Since tyrosinase exists in three oxidation forms each showing different ability to catalyze phenol or catechol substrates [26], the kinetics of the monophenolase activity is complicated and difficult to monitor [27]. Therefore, the monophenolase reaction was not determined in the current study.” (Line 121-126)
To study the potency of these peptides, the IC50 value could be determined for example with the mushroom enzyme.
Response: Thank you for the suggestion. The IC50 values of the three samples (GYSLGNWVCAAK, CEWHmono and CEWHdi) on mushroom tyrosinase were determined in our previously published papers. This statement is included in the Introduction:
- “According to the previous studies by our research group, CEW hydrolysates, e., 10 mg/mL of CEWHmono and CEWHdi inhibited 45.3 ± 0.66 monophenolase and 48.1 ± 0.23% diphenolase activities of mushroom tyrosinase, respectively [13].” (Line 60-62)
- The CEWHdi-derived peptide, GYSLGNWVCAAK, also gave an IC50 value of 3.04 ± 0.39 mM for tyrosinase diphenolase activity [16], which is more potent than that of FPY [17] and IQSPHFF [18] with IC50 values of 3.22 ± 0.9 and 4.0 mM, respectively. (Line 65-68)
For the potency of samples on mammalian tyrosinase, the IC50 values were not determined because the inhibition did not follow a dose-dependent manner. The highest inhibition of tyrosinase activity by GYSLGNWVCAAK and CEWHdi were 45.39 ± 1.31 and 50.89 ± 0.36%, respectively, at 10 µg/mL sample concentration. This statement is included in Section 2.3: “At 10 µg/mL, GYSLGNWVCAAK and CEWHdi gave the highest activity suppression (45.39 ± 1.31 and 50.89 ± 0.36%, respectively), which were higher than the positive control EGCG (39.65 ± 1.16%) at an equivalent concentration.” (Line 135-137).
Reviewer 3 Report
The manuscript describes the influence of chicken egg white peptide, chicken egg white-derived peptide and hydrolysates for ”wound recovery”.
I have some comments and question to the authors:
1. The title of the manuscript must be change. It does not reflect the contents of the article. In the article is discussed the skin pigmentation not wound healing.
2. Could you please more specify in the introduction the Fitzpatrick type III-VI based on the melanin content?
3. The cell line (B16F10) that you use in your experiment which skin type it is? In comparison to melanin.
4. Did you provide any microscopic evaluation of the cells (as in Fig 1) after application of chicken egg white peptide, chicken egg white-derived peptide and hydrolysates?
5. In the introduction I also missed the comparison with other natural products that could be used.
6. Is there any other effect of tyrosinase (except of influence on melanin) during wound healing?
7. In Fig 4 is missing caption (c) and (d) above second line of graph.
Author Response
The manuscript describes the influence of chicken egg white peptide, chicken egg white-derived peptide and hydrolysates for ”wound recovery”.
I have some comments and question to the authors:
- The title of the manuscript must be change. It does not reflect the contents of the article. In the article is discussed the skin pigmentation not wound healing.
Response: Thank you for the suggestion. The manuscript title has been revised to “Effect of chicken egg white-derived peptide and hydrolysates on abnormal skin pigmentation during wound recovery” (Line 2-3)
- Could you please more specify in the introduction the Fitzpatrick type III-VI based on the melanin content?
Response: The Fitzpatrick skin type is based on the melanin content (skin colour) and the skin response to UV exposures. The skin colour has been added to the Introduction: “It could happen to all skin types but tend to affect dark skinned individuals (Fitzpatrick type IV, V and VI, where the skin colour corresponds to olive, brown and black/dark brown, respectively) with greater frequency and severity [1].” (Line 33-36).
- The cell line (B16F10) that you use in your experiment which skin type it is? In comparison to melanin.
Response: B16F10 is a melanoma cell line derived from mouse (Mus musculus) which is used for all experiments conducted in the current study. It is a common cell line used for pigmentation-related studies [12]. Moreover, human and mouse tyrosinases are highly homologous with 86 and 84% identity at protein and cDNA levels, respectively [19].
[12] Song, Y.; Chen, S.; Li, L.; Zeng, Y.; Hu, X. The hypopigmentation mechanism of tyrosinase inhibitory peptides derived from food proteins: an overview. Molecules 2022, 27, 2710.
[19] Seruggia, D.; Josa, S.; Fernández, A.; Montoliu, L The structure and function of the mouse tyrosinase locus. Pigment Cell Melanoma Res.2021, 34, 212-221.
- Did you provide any microscopic evaluation of the cells (as in Fig 1) after application of chicken egg white peptide, chicken egg white-derived peptide and hydrolysates?
Response: Unfortunately, the microscopic evaluation of cells after application of samples was conducted due to the high cell viability (refer to Fig 2). However, we quantitatively examined the intracellular melanin level as shown in Fig. 4. We hope the reviewer can accept the data as it is.
- In the introduction I also missed the comparison with other natural products that could be used.
Response: Comparison of results with other natural products has been added to the Introduction: “To compare with other reports, the collagen hydrolysate showed 15.44 – 30.20% tyrosinase inhibition at 5 mg/mL [14]. The jellyfish hydrolysates showed 50% inhibition of tyrosinase activity at 14.1 – 24.5 mg/mL [15]. The CEWHdi-derived peptide, GYSLGNWVCAAK, also gave an IC50 value of 3.04 ± 0.39 mM for tyrosinase diphenolase activity [16], which is more potent than that of FPY [17] and IQSPHFF [18] with IC50 values of 3.22 ± 0.9 and 4.0 mM, respectively.” (Line 62-68)
- Is there any other effect of tyrosinase (except of influence on melanin) during wound healing?
Response: Based on our knowledge and literature search, tyrosinase plays no direct role in wound healing. It is the major enzyme for skin pigmentation. However, darkening of the wound is a common disorder following cutaneous injury due to hyperpigmentation (induced by hyperactivity of tyrosinase). Therefore, tyrosinase serves as the link between wound healing and the associated abnormal pigmentation in the current study. Inhibition of tyrosinase was proposed as an approach to alleviate hyperpigmentation during wound healing.
- In Fig 4 is missing caption (c) and (d) above second line of graph.
Response: Thank you for checking. The captions (c) and (d) have been added to Figure 4 on page 7.
Round 2
Reviewer 3 Report
Dear authors, thank you for your corrections. I think now it OK.